# Availability and Accessibility of Hydrography and Hydrogeology Spatial Data in Europe through INSPIRE

Danko Markovinović [1,*] , Vlado Cetl [1,*] , Sanja Šamanović [1] and Olga Bjelotomić Oršulić [2]

1   Department of Geodesy and Geomatics, University North, 42000 Varaždin, Croatia; sanja.samanovic@unin.hr
2   IGEA Ltd., 42000 Varaždin, Croatia; olga.bjelotomicorsulic@igea.hr
*   Correspondence: danko.markovinovic@unin.hr (D.M.); vlado.cetl@unin.hr (V.C.)

**Abstract:** Hydrography and hydrogeology data contain spatial references and as such are part of spatial data infrastructure. On the European level, these data are part of European spatial data infrastructure, well known as INSPIRE. The objective of INSPIRE is to make public spatial data available and accessible for a broad range of users in a simple, interoperable, and efficient way. Spatial data play an important role in facilitating data integration, enabling data-driven decision making on where and why things happen and easing communication through intuitive visualizations. Within this paper, we take the opportunity to reflect on the development and implementation of INSPIRE, with the main focus on the availability and accessibility of hydrography and hydrogeology data. By availability, we aim for the existence of metadata describing spatial data, while by accessibility, we aim for the existence of related services for spatial data viewing and downloading. The overall findings, based on the analysis in the INSPIRE Geoportal, shows that the data are still not fully available, although the deadline for INSPIRE implementation has already passed. Data accessibility is also an issue. Data that are even available in the infrastructure are sometimes not accessible. However, technological developments and recent policy initiatives could be drivers for future improvement.

**Keywords:** European spatial data infrastructure; INSPIRE; availability; accessibility; hydrography; hydrogeology

## 1. Introduction

Water is one of the most important resources in today's urban world. Information on the generation and availability of water resources is crucial for each country. Having reliable and authoritative data sources about water quality, the position of water resources, and the overall situations in relation to environmental protection are crucial factors for public, private, and other bodies that use this resource. Water resource management depends on the political situation and strategic guidelines, as well as on business and professional expectations. Management is a great challenge but also an obligation.

The availability and accessibility of public spatial data in European Union (EU) member states (MS) have always been issues in the past [1,2]. These included, e.g., different data policies, encodings, formats and semantics, licenses, pricing, etc. Data were also collected for, and applied to, domain specific use cases; comprehensive standards did not exist, impacting the reusability of such public sector data. The first attempts to improve this situation came with the 1994 presidential 'Executive Order 12906' [3], launching the US Spatial Data Infrastructure (SDI) initiatives. Based on it, the first ideas for a European SDI were put forward from 1995 onward under the 'GI 2000' umbrella [4]. While GI 2000 did not succeed with substantial operational steps towards a European SDI, it certainly created a favorable environment agreeing on the need for a common GI strategy and then infrastructure. In addition, there were other initiatives e.g., GIS of the Commission (GISCO), the public sector information (PSI) directive, CORINE land cover, etc. These initiatives intended to overcome key barriers affecting Europe in spite of the progress in SDI developments, which led to the creation of Infrastructure for Spatial Information in Europe



(INSPIRE). Following three years of intensive consultation among the EU member states and their experts, a public consultation, and the assessment of the likely impacts, the European Commission adopted the INSPIRE proposal for a directive in July 2004. Finally, the INSPIRE directive was adopted in 2007 [5], with the aim to directly address and solve the above mentioned set of problems.

The overall objective of the directive is to establish a European Union spatial data infrastructure (SDI) for the purposes of EU's environmental policies and policies or activities that have an impact on the environment. Its scope applies to spatial data held by, or on behalf of, public administration in performance of public tasks. INSPIRE is based on infrastructure for spatial information established and operated by EU MS and EFTA countries. Moreover, some non-EU countries, e.g., Western Balkan countries, Moldova, and the Ukraine are following INSPIRE principles [6]. The directive has 34 spatial data themes, under which datasets are categorized. The directive came into force on 15 May 2007; full implementation in every member state was required by 2021. It combines both a legal and a technical framework to make relevant spatial data available, accessible, and reused [7]. The vision for a European SDI has not changed since the inception of the directive, which is to promote cross-border data sharing and put in place easy-to-use, transparent, interoperable spatial data services, which are used in the daily work of environmental and other policy makers and policy implementers across the EU, at all levels of governance as well as businesses, science, and citizens [8,9].

As the deadlines for the INSPIRE implementation were in 2021, implementation of the directive is formally complete. So, it is a good moment to investigate how much INSPIRE infrastructure fulfilled the goal. How much data are available and accessible? Is it in accordance with FAIR (findable, accessible, interoperable, and reusable) data principles? In this paper, we focus on the availability and accessibility of hydrography and hydrogeology data. The INSPIRE directive provides a strong incentive and support for the development of water management policies and methods. Regardless of the fact that INSPIRE is technically complete, its implementation is the responsibility of all responsible bodies, at the state and local levels. The path to the implementation of the INSPIRE directive and care, in this case, for water resources, is neither simple nor easy and it requires full commitment to politics, profession, and science. It is a long-term process, so it is important to continuously monitor and indicate the state of geospatial data associated with these resources.

The paper is organized as follows. Following the introduction, we provide an overview of the INSPIRE infrastructure, including data scope, architecture, Geoportal, monitoring, and reporting. The third section investigates and analyzes availability and accessibility of hydrography and hydrogeology data based on the INSPIRE Geoportal. The fourth section discusses the results, summarizes the paper's conclusions, and offers possible directions for future research.

## 2. INSPIRE

Looking back to the 1990s, the availability and accessibility of public sector spatial data across Europe were minimal. Finding content was very difficult. Documentation was poor or missing and data were kept in incompatible formats. It was difficult and time consuming to find and combine datasets from different sources [2]. Data-sharing was also hampered by cultural and institutional barriers, including closed or non-existent data policies. On the other hand, spatial data were desperately needed. Climate change, natural disasters, such as floods, air pollution, or any environmental phenomenon do not stop at political borders [10,11]. In order to overcome these challenges, strong coordination was needed between stakeholders at both European and national levels. The most appealing solution for all was a creation of pan-European spatial data infrastructure (SDI), leveraging on existing national and regional data infrastructures. It was addressed by INSPIRE.

The legal framework for INSPIRE was set by the directive and related interdependent legal acts, which are called implementing rules, in the form of commission regulations and decisions. By design, the INSPIRE infrastructure is built upon national spatial data

infrastructures (NSDIs) established and operated by the EU MS and EFTA countries, made compliant with the implementing rules, covering its core components: metadata, network services, interoperability of spatial datasets and services, data sharing, and monitoring and reporting [12,13].

*2.1. Data Scope*

Hydrography data are part of core spatial data in Annex I, while hydrogeology data are part of data theme geology in Annex II (Figure 1).

The theme hydrography covers hydrographic elements, including marine areas and all other water bodies and items related to them, such as river basins and sub-basins. Geographically, it covers all inland water and marine areas covered by river basin districts as defined by the water framework directive (2000/60/EC) [14].

The hydrography theme is extensive and, therefore, has been broken into three application schema: base, network, and physical waters (Figure 2).

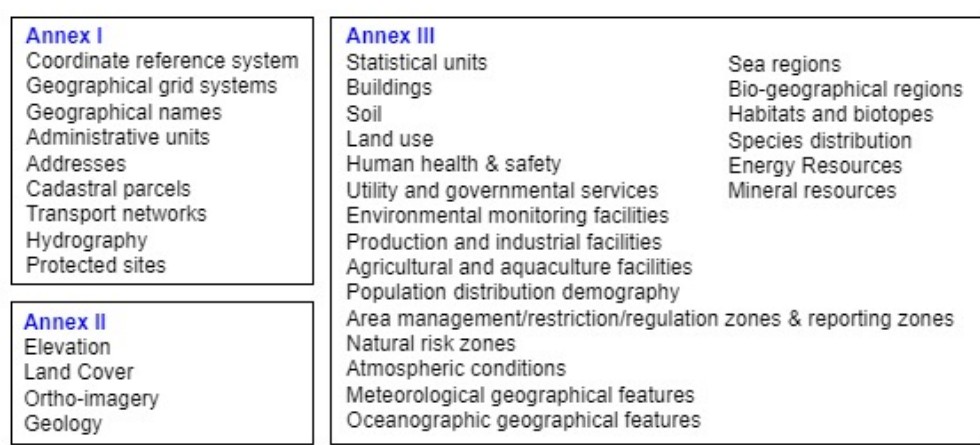

**Figure 1.** INSPIRE data themes, organized in three annexes. Source: INSPIRE directive (2007/2/EC) [5].

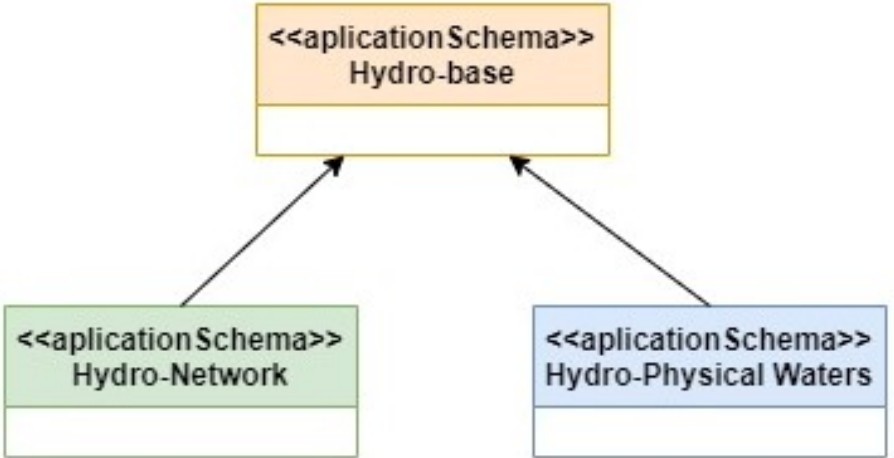

**Figure 2.** Hydrography package structure. Source INSPIRE consolidated UML Model (figure source [15]).

The hydro–base application schema provides a foundation for defining different views of hydrography. While there is only a single real world of hydrographic objects, it may have many representations. The INSPIRE hydrography theme identifies mapping, network, and reporting views as different representations of the real world, with three corresponding

application schema: base, network and physical waters [16]. Figure 3 illustrates how the real-world is modeled through the hydrography (HY) data model.

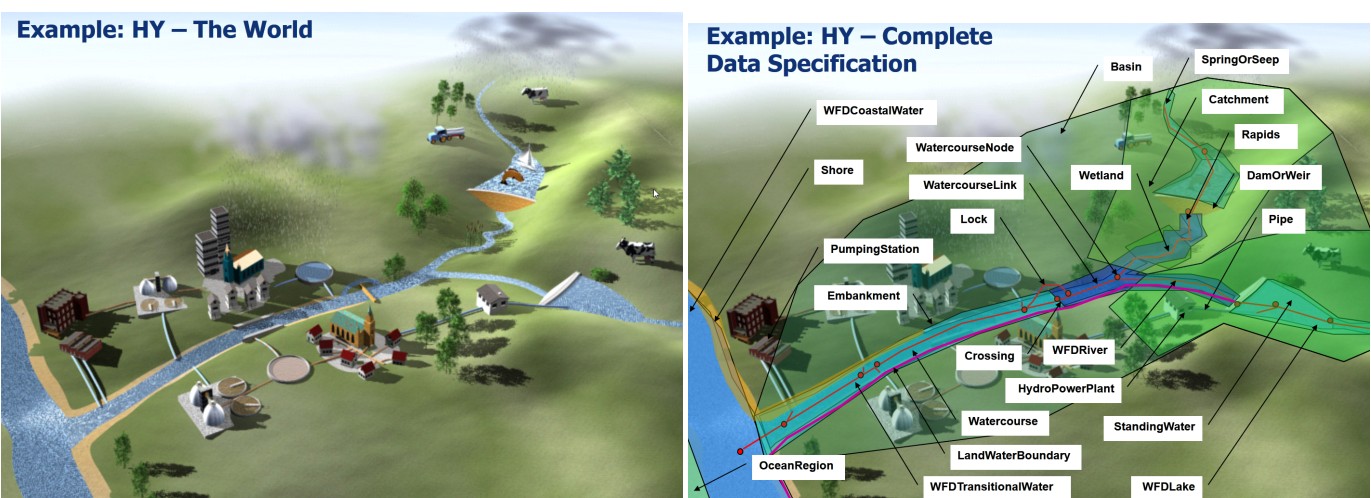

**Figure 3.** Abstraction of the real world in INSPIRE. Example from the "hydrography" data theme. Source: INSPIRE hydrography data Specification (figure source [16]).

Hydrogeology in INSPIRE is a part of the geology data theme [17]. Geology data specification defines three application schema: geology, hydrogeology, and geophysics, to provide the basic geological, hydrogeological, and geophysical knowledge on an area, with agreed sets of attributes.

Hydrogeology describes the flow, occurrence, and behavior of water in the underground environment. It is a science located between hydrology and geology, and both have a strong influence on the understanding of groundwater flow and solute transport. Hydrological processes are responsible, for example, for the characterization and understanding of the water supply derived from the recharge of aquifers. On the other hand, the physical properties and composition of the geologic materials (rocks and sediments) create the main environment for groundwater flow and storage. Rocks and sediments also influence groundwater quality in terms of their chemical composition.

The hydrogeological data model [18] contains:

- The aquifer system, comprising hydrogeologic units, aquifers, aquitards, aquicludes, and the aquifer system;
- The groundwater system, comprising groundwater body, and its relationships to the aquifer system, hydrogeology objects, and WFD groundwater body;
- Hydrogeology objects, both natural and man-made, including wells.

### 2.2. INSPIRE Architecture

Data, functionality, and metadata are shared through web-based services, referred to as network services [19], based on a service oriented architecture (SOA) approach, as shown in Figure 4.

Datasets are distributed among public organizations. In order to make them findable, it is necessary to create metadata that are served through discovery services. This also includes service metadata. Both types of metadata are crucial for users in order to find data and services and to assess their fit for purpose. Once when data are found, users can use different types of network services to access the data (i.e., view, download, etc.). Data and network services are a crucial part of the infrastructure.

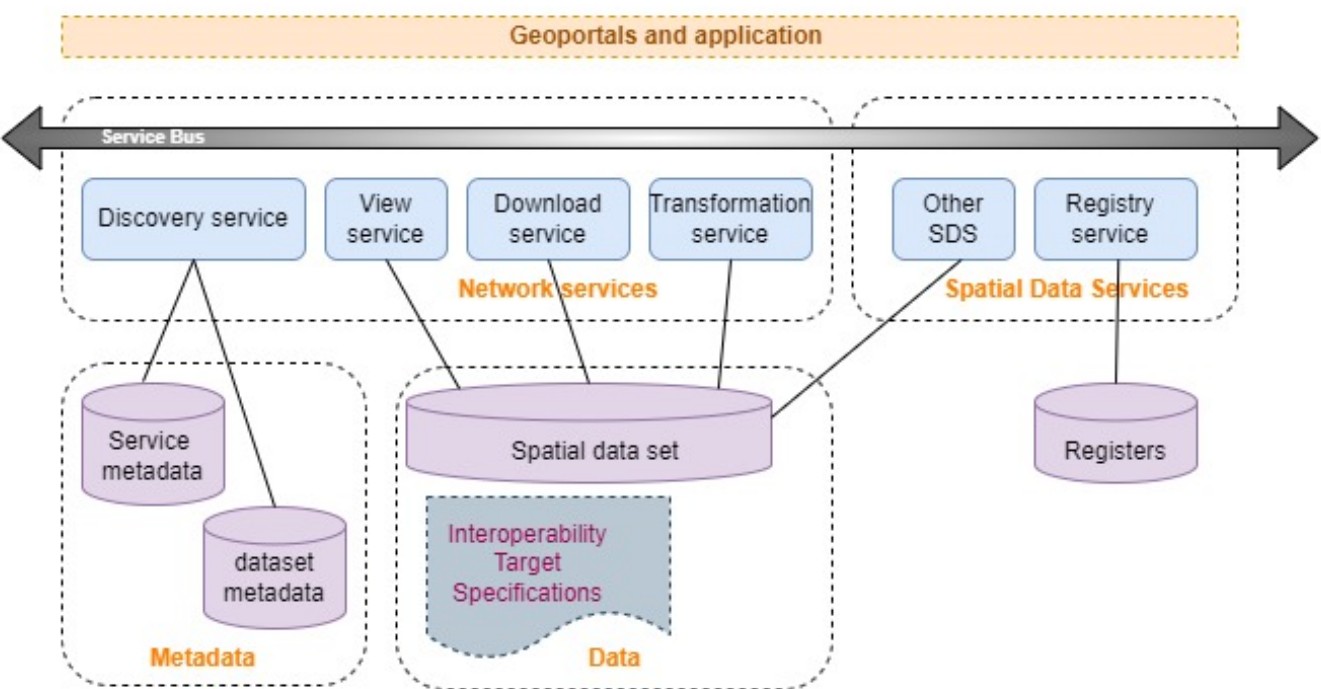

**Figure 4.** Distributed service oriented architecture of INSPIRE. Source: European Commission, Joint Research Centre. (figure source [7]).

Network services are implemented through well-established international standards, mainly developed by the Open Geospatial Consortium (OGC) [20]. Technical guidance documents illustrate how data providers establish access to metadata for discovery services through the Catalogue Service for the Web (CSW). Similar to view services, the interactive visualization of geo-referenced content involves guidance using the web map service (WMS) and web map tile service (WMTS) standards. Download services also have guidelines that recommend the use of Atom feeds, web feature service (WFS), web coverage service (WCS), and sensor observation service (SOS), for appropriate types of data. There are also various transformation services defined, which can support coordinate and data transformations. In addition to all of the above, there are generic services (registry and other spatial data services), which are implemented on a national as well as European level.

### 2.3. Implementation Roadmap

The directive came into force on 15 May 2007; full implementation of every member state was required by 2021. Full implementation implied provision of metadata, spatial data, and spatial data services for all data themes listed in Annex I, II, and III.

The deadline for member states to transpose the directive into their national legislation was 15 May 2009. The implementation process started immediately after, following an implementation roadmap that treated each component individually through a stepwise approach [21]. The first important milestone was in December 2013, when member states were obliged to provide their data 'as-is'. This step established metadata and exposed data through network services. Consequently, by December 2017, datasets that fell under the scope of Annex I were expected to be in place and interoperable. Similarly, by the end of 2020, data for Annex II and III should have also been in conformance with the directive's requirements.

The results of the INSPIRE 2019 and 2020 monitoring and reporting show that the status of implementation of INSPIRE is heterogeneous across countries, and there is no single country that has yet achieved full implementation according to the INSPIRE roadmap [9].

## 2.4. INSPIRE Geoportal

The entry point to the INSPIRE infrastructure is the EU Geoportal [13,22], depicted at the top of the INSPIRE architecture in Figure 4. It serves as a central access point to the data and services from public organizations in the EU member states (MS) and European Free Trade Association (EFTA) countries that fall under the scope of INSPIRE. The INSPIRE Geoportal enables cross-border data discovery, visualization, and downloads, in addition to metadata, data, and service validation. The Geoportal does not store any geospatial data, it simply acts as the main client application of the whole INSPIRE infrastructure by exposing data through the harvesting of the CSW endpoints made available by MS. Alongside the INSPIRE Geoportal, which is operated by the European Commission, there are also national geoportals operated by single countries. Links to national geoportals are available in the INSPIRE knowledge base (IKB) section: INSPIRE in your country [23].

The Geoportal is a one-stop shop for public authorities, businesses, and citizens to access and use geospatial datasets related to the environment in Europe (Figure 5). It also provides overviews of the availability of datasets by country and thematic area, and provides ready-to-use data either through interoperable services or by direct download, to maximize their exploitation in third-party GIS clients and applications.

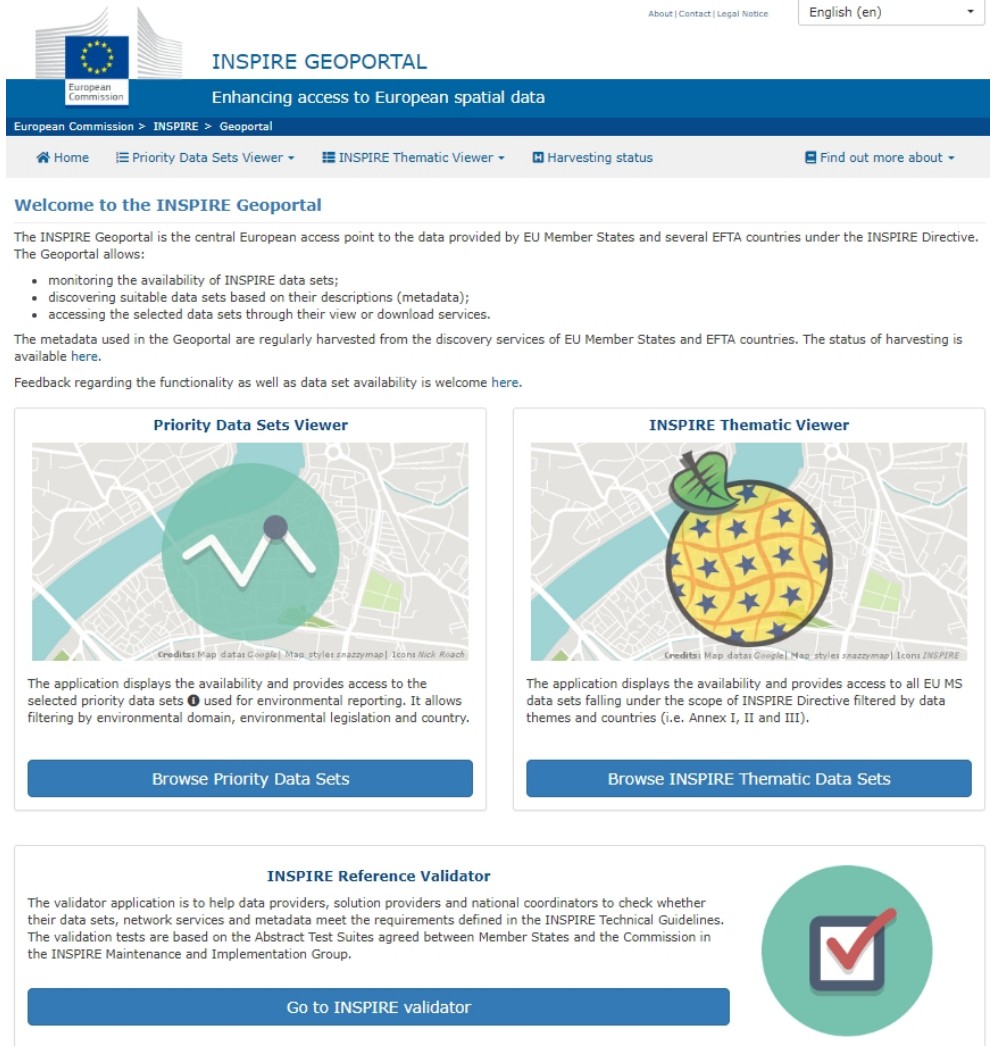

**Figure 5.** Landing page of the INSPIRE Geoportal. Source: European Commission, Joint Research Centre.

The Geoportal consists of three main applications:

1.  Priority datasets Viewer, which displays the availability and provides access to the priority datasets used for environmental reporting;
2.  INSPIRE Thematic Viewer, which displays the availability and provides access to all EU MS and EFTA datasets falling under the scope of the INSPIRE directive, filtered by data themes and/or countries;
3.  INSPIRE Reference Validator, which helps data providers check whether their datasets, services, and metadata meet the INSPIRE requirements.

The input source to the INSPIRE Geoportal is the harvesting of metadata from the officially registered Discovery Services of EU MS and some EFTA countries. Currently 36 Discovery Services are harvested on a regular basis and information is available under the harvesting status section of the Geoportal.

Insight into the current status of the infrastructure is provided by the INSPIRE Thematic Viewer, which offers two possibilities for browsing datasets: by individual EU MS and EFTA countries and by INSPIRE data theme. Figure 6 shows the availability of hydrography datasets in EU MS and EFTA countries as of 3 March 2022.

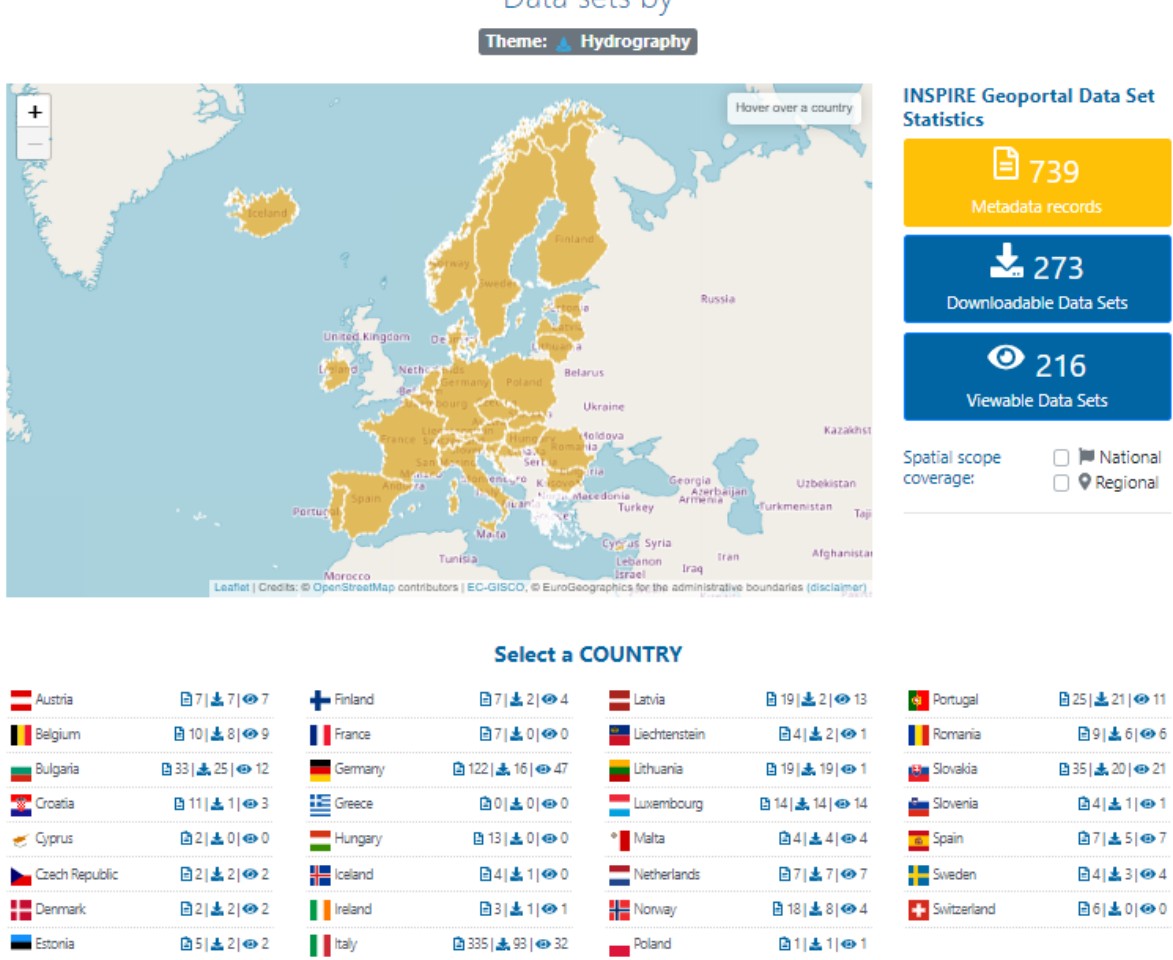

**Figure 6.** Available hydrography datasets in EU MS and EFTA countries. Source: European Commission, Joint Research Centre.

The figure shows the total numbers of metadata records (739), downloadable datasets (273), and viewable datasets (216). It is also possible to filter search results in accordance with spatial coverage (regional or national). At the bottom of the page there is an overview for each MS and EFTA country with individual country results. Again, the first number shows metadata records, the second downloadable datasets and the third is the number of

viewable datasets. Click on a particular country, e.g., Austria (Figure 7), provides a more detailed overview for that country.

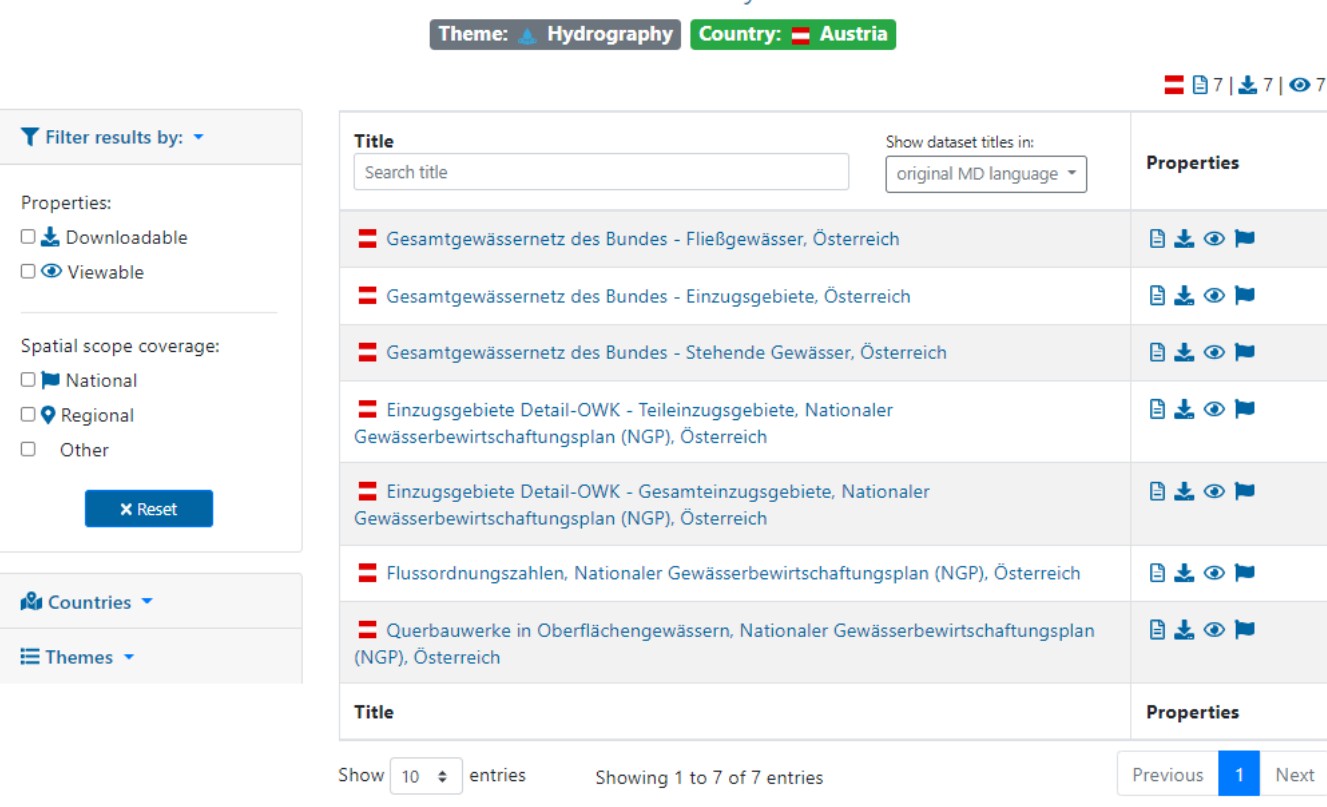

**Figure 7.** Available hydrography datasets in Austria. Source: European Commission, Joint Research Centre.

Metadata records are usually provided in the national language (e.g., German), but could be automatically translated to English. Click on a chosen metadata record provides detailed metadata about the resource together with related links to view and download services if they exist.

When browsing datasets by the INSPIRE data theme in the Thematic Viewer, it is not possible to perform a more detailed search. The ultimate level is the data theme. Since hydrogeology is a part of the geology data theme, in order to search for these data, it is necessary to use the resource browser, which is available under the 'find out more about' section in Metadata Tools (Figure 8).

The INSPIRE Metadata Browser allows a more detailed search, basically by each metadata element available. The most simple is to use keywords (e.g., hydrogeology). It is possible then to add more keywords and combine them in order to perform a deeper search (e.g., hydrogeology plus country name). The Metadata Browser is certainly not as user friendly as the Thematic Viewer, but offers much more search possibilities.

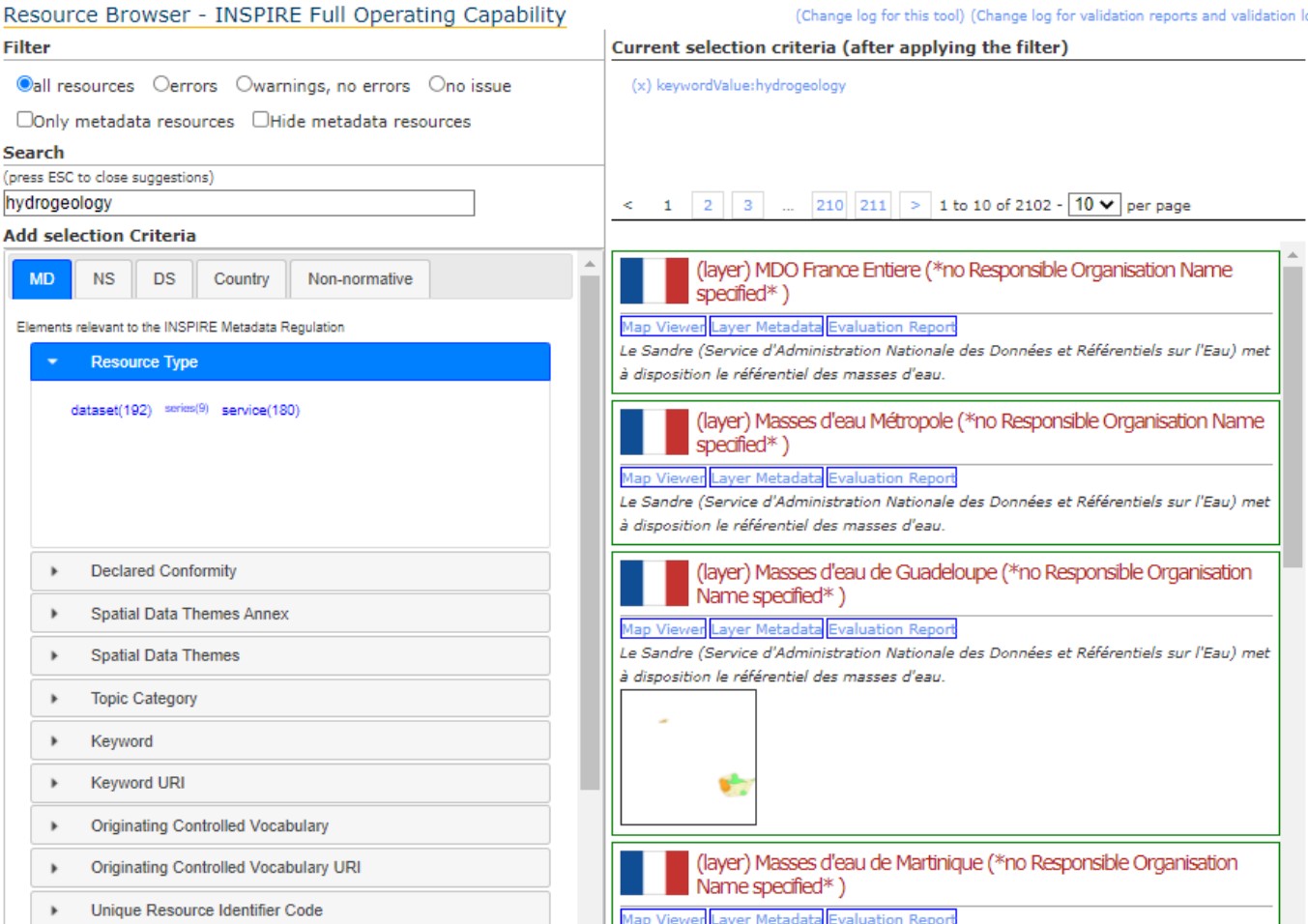

**Figure 8.** INSPIRE Metadata Browser, Source: European Commission, Joint Research Centre.

*2.5. INSPIRE Monitoring and Reporting*

The INSPIRE monitoring and reporting is an annual process, whose results are published by 31 March each year, at the latest, with reference to the status of the implementation of the infrastructure on 15 December of the preceding year. Monitoring is performed through a set of indicators that are calculated based on the metadata collected from MS and EFTA countries' public authorities. These indicators measure the implementation progress of the directive and are used to evaluate the success against its objectives. For the reporting, countries must provide the relevant information in the INSPIRE knowledge base when changes occur in the governance of the national SDIs as part of the country fiche. The automated calculation of 19 indicators through the direct use of the INSPIRE Geoportal and INSPIRE Reference Validator were firstly introduced in 2019 by Commission Implementing Decision (EU) 2019/1372 [24].

There are 19 monitoring indicators grouped into five categories:

- Monitoring of the availability of spatial data and services;
- Monitoring of the conformity of metadata;
- Monitoring of the conformity of spatial datasets;
- Monitoring of the accessibility of spatial datasets through view and download services;
- Monitoring of the conformity of network services.

The results of annual monitoring are published every year in the INSPIRE knowledge base as part of the country fiches. Monitoring does not look on individual data themes but on the total number of metadata records in accordance with three annexes.

For the purpose of our research, we focused on indicators related to availability and accessibility of spatial datasets through view and download services for hydrography and hydrogeology data. Namely, we chose following indicators:

- DSi1.1—the number of spatial datasets for which metadata exist. The indicator represents the amount of all dataset metadata records, published by MS and EFTA countries through their discovery services.
- DSi1.2—the number of spatial data services for which metadata exist. This is the number of spatial data services published by MS and EFTA countries through their discovery services.
- DSi1.4—the number of spatial datasets that cover regional territory. This is the number of spatial dataset metadata records that contain a keyword "Regional".
- DSi1.5—the number of spatial datasets that cover the national territory. This is the number of spatial dataset metadata records that contain a keyword "National".
- NSi2.1—the percentage of spatial datasets that are accessible through view services. This is the number of spatial datasets for which a view service exists, multiplied by a hundred and divided by the number of spatial datasets as given by indicator DSi1.1.
- NSi2.2—the percentage of spatial datasets that are accessible through download services. This is the number of spatial datasets for which a download service exists, multiplied by a hundred and divided by the number of spatial datasets as given by indicator DSi1.1.

## 3. Data Analysis

In this analysis, two water-related INSPIRE thematic areas were observed: hydrography and hydrogeology. Hydrography is one of nine themes from the Annex I; hydrogeology is the sub-theme within the geology, one of four themes from the Annex II. To assess the availability and accessibility of the data, we followed the quantitative indicators used in monitoring and reporting for all member countries. These values refer to the possibility to access, download, or view some dataset, service, or metadata directly from the INSPIRE Geoportal. The values used in this analysis are compliant with the commission decision (EU) 2019/1372 [24], from which we used five of them, given in the following section.

### 3.1. Hydrography

In the Annex I thematic areas, hydrography is among the areas with the most data included with INSPIRE datasets and services, covering hydrographic elements, marine and lake areas, river and sub-river basins, and other water bodies.

The overall number of metadata records regarding the hydrography thematic area shows 739 records. Although it seems quite a large number of available records, keeping in mind all of the EU and several EFTA countries involved in the INSPIRE area, we noticed significant differences between the countries: Germany and Italy are the only ones having more than 100 records each, jointly making a share of more than 61% of all records in the hydrography group (see Table 1, the DSi1.1 indicator). Other shares move between the few to 35 records, as maximum. The country without any records is Greece; Switzerland and Cyprus did declare to have a few meta records, but do not offer any dataset or service to download or view. In the following tables and graphs, for the abbreviated country names, we use standard ISO Alpha2 codes (https://www.iso.org/obp/ui/#search (accessed on 10 February 2022).

The DSi1.1-DSi1.5 indicators are absolute values showing the exact number of records held by each country on the list. The numbers were obtained through the INSPIRE Geoportal in the period of 10 February–5 March 2022. During the data-gathering period, we noticed the dynamics of the data: the declared number of available data are changing almost daily with overall change in our observing period up to 10% in summed values for all the countries. This variation is expected but still has to be addressed, since the analysis of such vivid datasets has to be attached for a period in time when it is observed and analysis can be done only temporarily. The INSPIRE services and datasets are subject

to constant upgrade and harmonization made by its main providers and, therefore, are constantly oscillating in a number of records. The DSi1.1, as seen in Table 1, is the number of spatial datasets for which metadata exist and it is by far the greatest number in the table. As for the opposite, the downloadable and viewable sets, which are the main data services used by the average user, are left far behind by their indicators: the summed value of both spatial data services for which metadata exist is 501, while DSi1.1 is 739, for the 40% higher.

**Table 1.** Indicators of availability for hydrography.

| Country | DSi1.1 | Downloadable | Viewable | DSi1.2 | DSi1.4 | DSi1.5 | NSi2.1 | NSi2.2 |
|---------|--------|--------------|----------|--------|--------|--------|--------|--------|
| AT | 7 | 7 | 7 | 14 | 0 | 7 | 100.0% | 100.0% |
| BE | 10 | 8 | 9 | 17 | 7 | 3 | 90.0% | 80.0% |
| BG | 33 | 25 | 24 | 49 | 0 | 25 | 72.7% | 75.8% |
| CH | 6 | 0 | 0 | 0 | 0 | 0 | 0.0% | 0.0% |
| CY | 2 | 0 | 0 | 0 | 0 | 0 | 0.0% | 0.0% |
| CZ | 2 | 2 | 2 | 4 | 0 | 2 | 100.0% | 100.0% |
| DE | 122 | 16 | 47 | 63 | 81 | 5 | 38.5% | 13.1% |
| DK | 2 | 2 | 2 | 4 | 0 | 2 | 100.0% | 100.0% |
| EE | 5 | 2 | 2 | 4 | 0 | 5 | 40.0% | 40.0% |
| EL | 0 | 0 | 0 | 0 | 0 | 0 | 0.0% | 0.0% |
| ES | 7 | 5 | 7 | 12 | 1 | 6 | 100.0% | 71.4% |
| FI | 7 | 2 | 4 | 6 | 0 | 3 | 57.1% | 28.6% |
| FR | 7 | 0 | 0 | 0 | 0 | 7 | 0.0% | 0.0% |
| HR | 11 | 1 | 3 | 4 | 0 | 10 | 27.3% | 9.1% |
| HU | 13 | 0 | 0 | 0 | 0 | 0 | 0.0% | 0.0% |
| IE | 3 | 1 | 1 | 2 | 0 | 3 | 33.3% | 33.3% |
| IS | 4 | 1 | 0 | 1 | 0 | 0 | 0.0% | 25.0% |
| IT | 335 | 93 | 32 | 125 | 270 | 8 | 9.6% | 27.8% |
| LI | 4 | 2 | 1 | 3 | 0 | 4 | 25.0% | 50.0% |
| LT | 19 | 19 | 1 | 20 | 0 | 19 | 5.3% | 100.0% |
| LU | 14 | 14 | 14 | 28 | 0 | 14 | 100.0% | 100.0% |
| LV | 19 | 2 | 13 | 15 | 0 | 0 | 68.4% | 10.5% |
| MT | 4 | 4 | 4 | 8 | 0 | 4 | 100.0% | 100.0% |
| NL | 7 | 7 | 7 | 14 | 0 | 6 | 100.0% | 100.0% |
| NO | 18 | 8 | 4 | 12 | 1 | 17 | 22.2% | 44.4% |
| PL | 1 | 1 | 1 | 2 | 0 | 1 | 100.0% | 100.0% |
| PT | 25 | 21 | 11 | 32 | 16 | 9 | 44.0% | 84.0% |
| RO | 9 | 6 | 6 | 12 | 2 | 7 | 66.7% | 66.7% |
| SE | 4 | 3 | 4 | 7 | 0 | 4 | 100.0% | 75.0% |
| SI | 4 | 1 | 1 | 2 | 0 | 4 | 25.0% | 25.0% |
| SK | 35 | 20 | 21 | 41 | 0 | 31 | 60.0% | 57.1% |
| overall | 739 | 273 | 228 | 501 | 378 | 206 | | |

As seen in Figure 9, the mutual relation between the DSi1.1 and the DSi1.2 indicator cannot be described as some predictive pattern. Concerning two of the countries with the largest number of data services available, Germany and Italy, the spatial dataset for which metadata exist (DSi1.1) are significantly larger than the number of the spatial data services with the metadata (DSi1.2), while there is a great number of other countries with this ration in favor of DSi1.2.

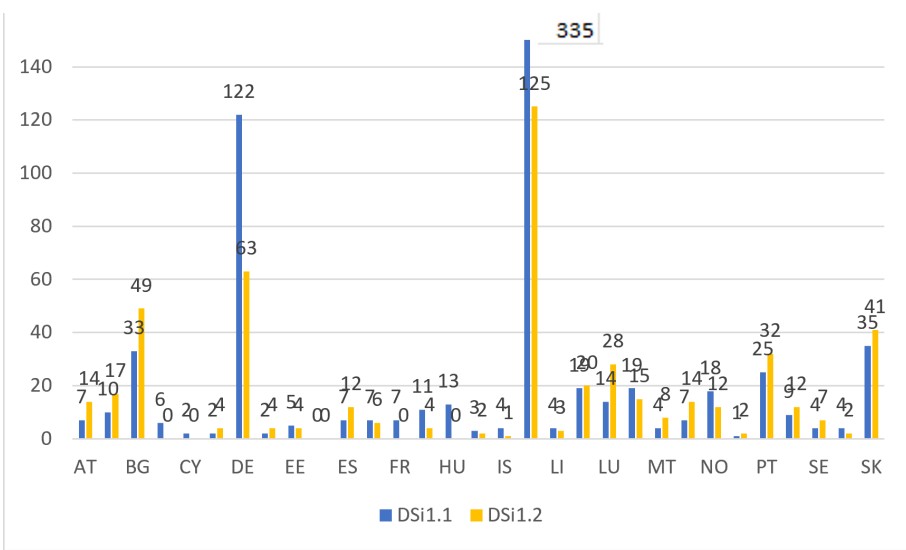

**Figure 9.** DSi1.1 and DSi1.2 indicators for the hydrography.

The DSi1.4 and the DSi1.5 are the indicators of the number of spatial datasets that cover regional and national territory, respectively. For the hydrography theme, it was possible to extract these indicators directly from INSPIRE Geoportal. For better insight into all values on the chart, the ordinate axis was limited to 100, despite Italy having 270 DSi1.4 indicator value. The green bar indicates the DSi1.4 national coverage, and the blue bars are indicators of the DSi1.5 regional coverage with spatial datasets. As seen from the Figure 10, most of the countries significantly 'prevail' the indicator of the national coverage, while only two countries with the biggest datasets generally available have the regional coverage indicator, the DSi1.5, significantly stronger than the national one.

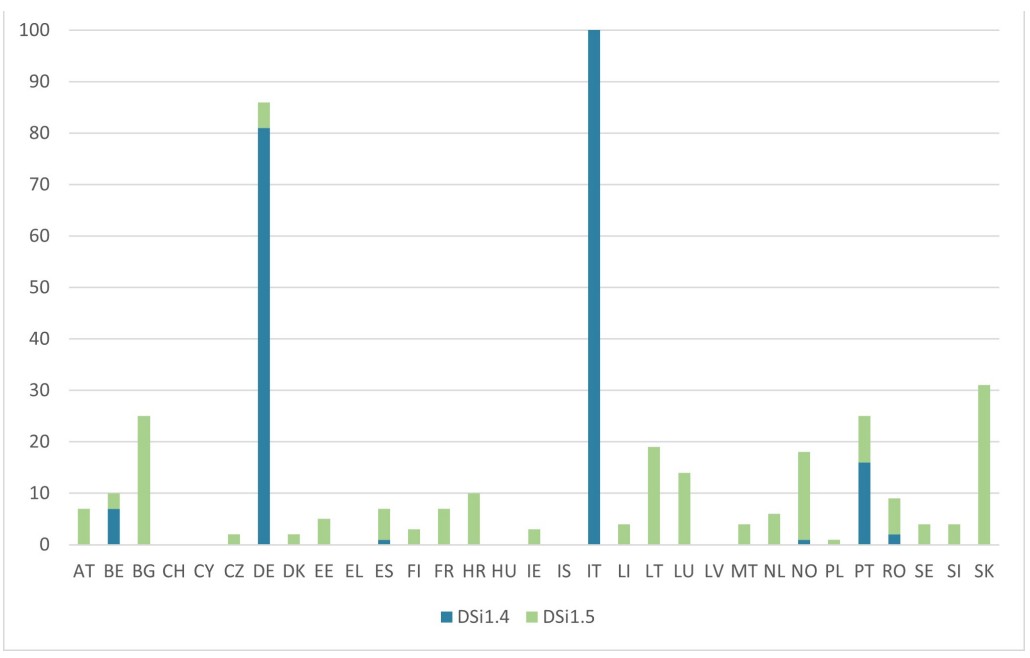

**Figure 10.** Indicators DSi1.4 and DSi1.5 for hydrography.

As for the NSi2.1 and NSi2.2 indicators that represent the percentage of spatial datasets that are accessible through view and download services, respectively, Figure 11 shows the high percentage in most of the countries. We noticed curiosity at Germany and Italy, which are the two countries with all other indicators significantly higher than others, but in this case, when it comes to the percentage of really downloadable and viewable data services

regarding the declared absolute numbers, their NSi2.1 and NSi2.2 indicators are by far the lower ones.

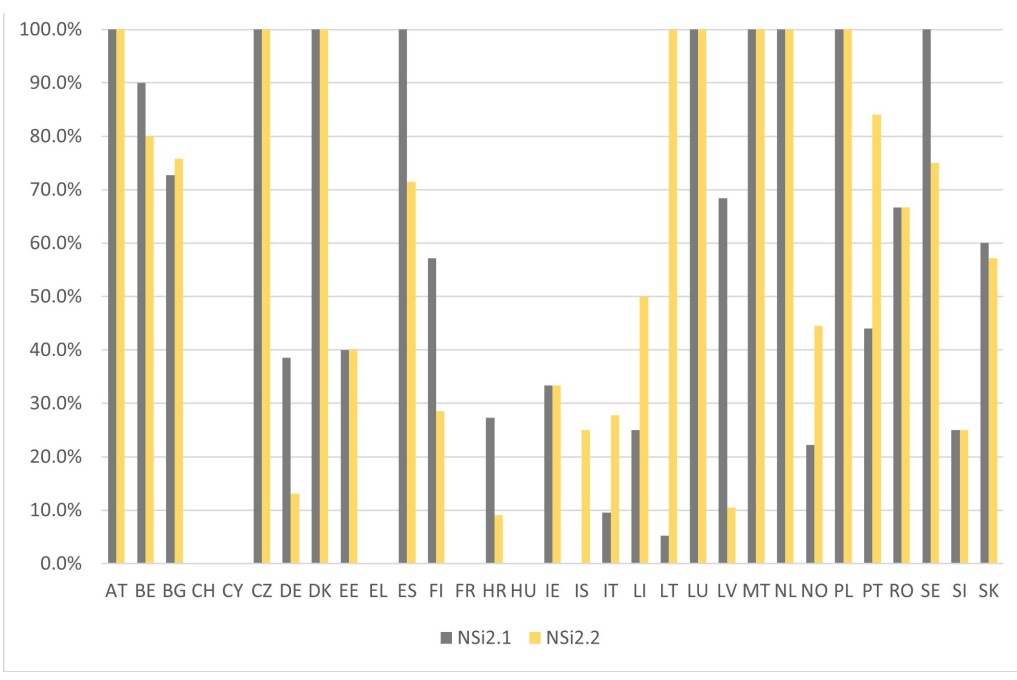

**Figure 11.** Indicators NSi2.1 and NSi2.2 for hydrography.

### 3.2. Hydrogeology

Hydrogeology is situated in Annex II, geology theme, which is the richest subject in the group regarding the included datasets and services. However, it has to be highlighted that hydrogeology can be determined only by assessment through the Resource Metadata Browser, the additional tool of the INSPIRE Geoportal, which allows a detailed search within a certain subject area, but has its limitations regarding the clear and undoubtable numbers of datasets according to the keyword we made the search about. As for the keyword "hydrogeology", 2102 datasets and data services were found. All the data mining were made in the period of 10 February to 5 March, 2022, whilst the oscillations in the overall number were not as vivid as noticed in the hydrography theme. The situation in the hydrogeology section is quite different: there is almost a three times larger number of overall records than in the hydrography section, but the number of countries providing the datasets and services is significantly smaller. As seen in Table 2, 12 of 30 countries do not offer any of the datasets of services regarding the hydrogeologic data. On the contrary, Belgium, Germany, and Italy provide hundreds of data records within this subject. In the Resource Metadata Browser, it is not possible to conduct research on DSi1.4 and DSi1.5, the indicators of the records regarding the national and regional coverage. Therefore, the two mentioned indicators were left out in the analysis of hydrogeology sources. Furthermore, the analysis is slightly different through the Resource Metadata Browser than it is through the country overview thematic viewer of the INSPIRE Geoportal: the filter is made with the keywords and through the pre-default sections: within the resource type, we obtained the number of datasets and services separately, and within the spatial data service type section, we could extract the number of downloadable and viewable data provided. These numbers were mandatory to compute the percentage of viewable and downloadable datasets and services regarding the declared values. The results are interesting: NSi2.1 and NSi2.2 indicators mainly varied between 25 and 30%, meaning that, in most cases, only 30% of the declared data services are really available to the user to be used, regardless if it is in downloadable or viewable manner. Moreover, the interesting numbers brought our attention to the following: in Belgium and Italy, the countries that have several hundreds of metadata records noted, below 1% of these data are available and accessible to the end user, while countries that

have a dozen or two of the metadata records declared regularly reach 30% of the data services accessible to the end user.

**Table 2.** Indicators of availability for hydrogeology.

| Country | DSi1.1 | Datasets | Services | D'loadable | Viewable | DSi1.2 | NSi2.1 | NSi2.2 |
|---|---|---|---|---|---|---|---|---|
| AT | 12 | 6 | 6 | 2 | 4 | 6 | 33.3% | 16.7% |
| BE | 747 | 31 | 3 | 2 | 1 | 3 | 0.1% | 0.3% |
| BG | | | | | | | | |
| CH | 3 | | | | | | | |
| CY | | | | | | | | |
| CZ | | 2 | | | | | | |
| DE | 108 | 20 | 61 | 29 | 32 | 61 | 29.6% | 26.9% |
| DK | | | | | | | | |
| EE | 20 | 10 | 8 | 4 | 4 | 8 | 20.0% | 20.0% |
| EL | | | | | | | | |
| ES | 21 | 5 | 14 | | 14 | 14 | 66.7% | 0.0% |
| FI | 4 | | 2 | | 2 | 2 | 50.0% | 0.0% |
| FR | 31 | 20 | 5 | | | | | |
| HR | | | | | | | | |
| HU | | | | | | | | |
| IE | 2 | 2 | | | | | | |
| IS | | | | | | | | |
| IT | 1032 | 44 | 36 | 17 | 19 | 36 | 1.8% | 1.6% |
| LI | | | | | | | | |
| LT | 5 | | | | | | | |
| LU | | | | | | | | |
| LV | 35 | 13 | 22 | 9 | 13 | 22 | 37.1% | 25.7% |
| MT | 2 | | | | | | | |
| NL | 13 | 4 | 5 | 3 | 2 | 5 | 15.4% | 23.1% |
| NO | 1 | 1 | | | | | | |
| PL | 7 | 2 | 4 | 2 | 2 | 4 | 28.6% | 28.6% |
| PT | 11 | 2 | 6 | 3 | 3 | 6 | 27.3% | 27.3% |
| RO | | | | | | | | |
| SE | 10 | | | | | | | |
| SI | 38 | 30 | 8 | 2 | 6 | 8 | 15.8% | 5.3% |
| SK | | | | | | | | |
| overall | 2102 | 192 | 180 | 73 | 102 | 175 | | |

The indicators DSi1.1 and DSi1.2 for hydrogeology are shown on Figure 12. The ordinate of the chart is scaled down to 120 as the maximum value, cutting the enormous values of Belgium and Italy, which extend to 747 and 1032, respectively, as given in Table 2, which would provoke the impression that other countries are around zero if the two countries are displayed in their full value.

As seen in Figure 10, the same as shown for the hydrography theme, the two of the countries with the largest number of data services available, Belgium and Italy, the spatial dataset for which metadata exist (DSi1.1), are significantly larger than the number of the spatial data services with the metadata (DSi1.2). On the other hand, unlikely as for the hydrography DSi1.1 and DSi1.2 relation, in the hydrogeology, most of the countries by default give significantly larger numbers of the DSi1.1 indicator than for the DSi1.2, implying that the spatial data services are not developed as much as spatial datasets in the hydrogeology group.

As for the NSi2.1 and NSi2.2 indicators that represent the percentage of spatial datasets that are accessible through view and download services, respectively, Figure 13 shows the status for hydrogeologic data through the INSPIRE Geoportal. Mostly, there is a high coincidence between the NSi2.1 and NSi2.2 indicators for each country, except for the two countries, namely Spain and France, which have the greatest NSi2.1 indicator, giving the

wide span of viewable services but lacking the possibility of downloadable services, which can be seen in Figure 13 where there is no visible bar for the NSi2.2 for these two countries.

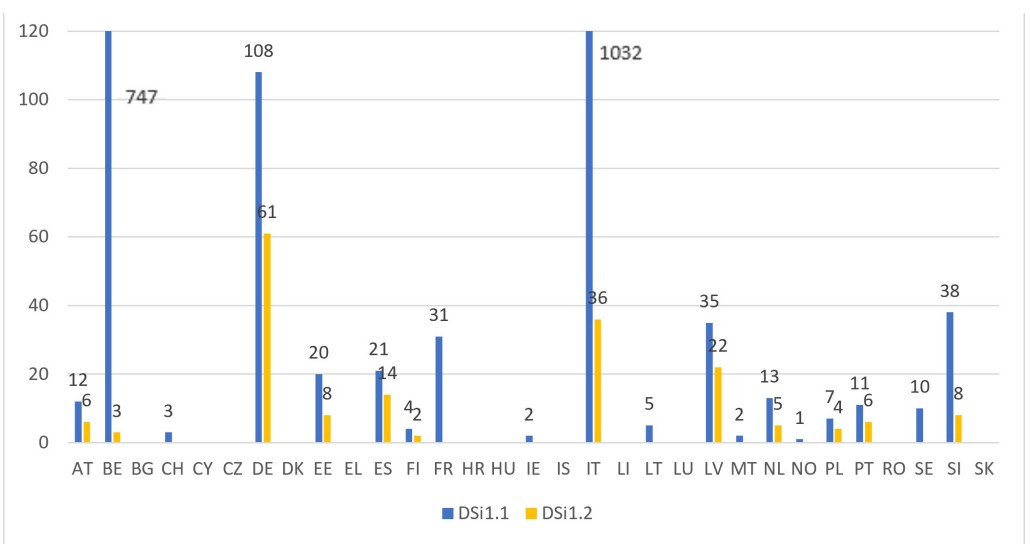

**Figure 12.** Indicators DSi1.1 and DSi1.2 for hydrogeology.

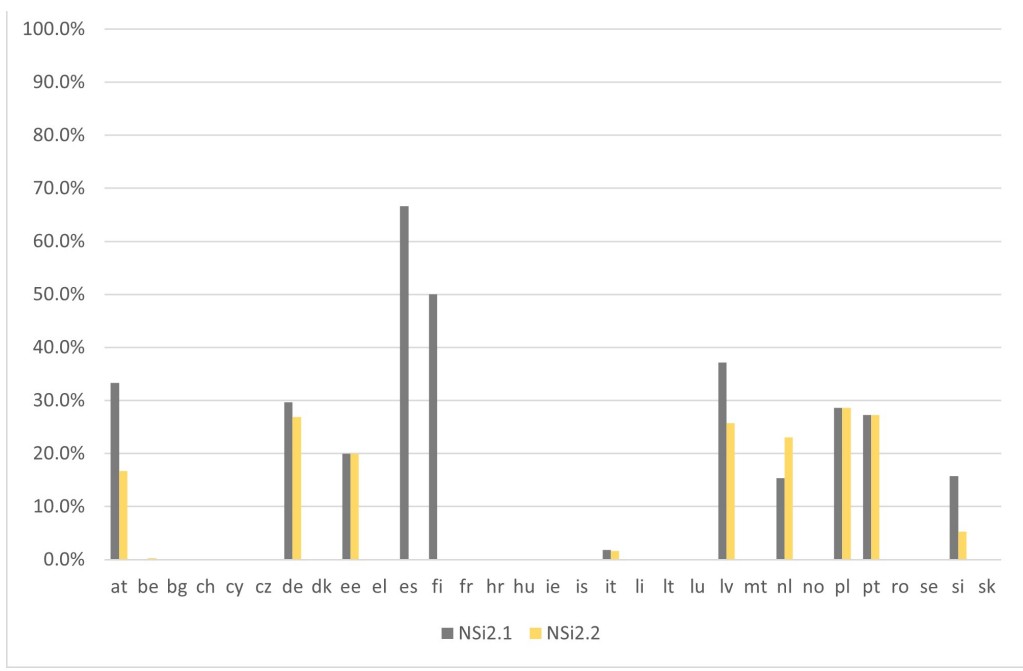

**Figure 13.** Indicators NSi2.1 and NSi2.2 for hydrogeology.

The indicators of availability and accessibility of spatial data are introduced by the Commission Implementing Decision for the purpose of quantitative evaluation of declared spatial datasets and services provided through national contact points of spatial data infrastructure and the INSPIRE Geoportal, consequently. Since the indicators are generated only based on declared descriptive data that each country provides, the information is not reliable, pertaining to really accessible and available data, and should be taken with caution. To evaluate the real percentage of available and accessible spatial datasets, we made a meticulous analysis over each declared dataset to obtain the exact number of really viewable and downloadable datasets, regarding the ones declared. As for the hydrography subject, the analysis was made over all 31 countries, the same shown in Tables 1 and 2, from which 8 countries declared zero downloadable datasets. The analysis resulted in 7 of 24 countries that declared some number of available datasets that did not have any data

available and accessible through their provided web links. The data are either declared as "page is not available" or "datasets cannot be accessed", and similar. Our assumption is that the services changed in the meantime from the time links placed on INSPIRE Geoportal, but nobody further updated the links or changed the service details. Furthermore, 16 remaining countries have more than 75% of datasets really available and accessible.

Overall analysis of really accessible and available datasets for the hydrography theme showed a very high percentage of accessible and available datasets in 52% of the analyzed countries. Unfortunately, for the remaining 48% of them, either self-declared zero available datasets or the page and links were not up to date.

New policies and directives about the national and European spatial data infrastructure have put data and data sharing ideas at the focus of the spatial data digital transformation in order to address the urgent social and environmental challenges and, consequently, create the basis for future strategies and regulations regarding our environment and its protection. Following the INSPIRE agenda and roadmap, very favourable conditions have been set up to take a close look at the INSPIRE development thus far, for critical assessment of the achievements and failures, and, to build on these insights, to define or redefine a vision, actions, and conditions to achieve it.

An analysis made in this paper on the available water-related INSPIRE data show that some countries do not have data-sharing services for hydrography and hydrogeology data. The unavailability of the data through the service is one of the fundamental problems in the implementation of the INSPIRE directive. Countries that do not have view and download services should first define the workflow for it. This includes the establishment of a database, preparation of IT infrastructure, data hosting sites, as well as the sharing arrangements or licenses according to the central point (institution) responsible for the implementation of the INSPIRE directive.

All of these steps require human and financial resources, knowledge and experience, legal frameworks, and implementation documentation, and depend on strategic state decisions. One of the main shortcomings observed in the multi-year monitoring of the INSPIRE directive, is the lack of human or professional capacity. Institutions responsible for the data should have professionals in place to take care of geospatial data. Professional capacities should be included in the business processes of collecting, analysing, processing, updating, and sharing geospatial data. Experts in addition to domain knowledge in geomatics, in terms of data collection with advanced geo sensors, should also have a broader knowledge in the field of analysis and statistical data processing. Therefore, continuous training of experts in the form of participation in scientific conferences, workshops, and meetings related to the INSPIRE directive is recommended.

Furthermore, there are countries that have initially started the INSPIRE process and declared the data services and download possibilities on the main INSPIRE Geoportal, but the service connections have not been updated or the data have never been shared. The reasons can only be assumed: the sluggishness and ignorance of the state public institutions on the need for further development and care of such shared services and data, the inner policies and regulations and their obsolete restrictions, the budget issues that prevent the data and services to be harmonized and completely digitally prepared for the public announcement, and so on. For these countries, we recommend more promotion from national contact points (NCPs), which are national bodies responsible for the INSPIRE implementation, more reminders and workshops about the know-how for the service formats, their volume and the roadmap to follow with the harmonization during the time. We noted this issue that occurs in the countries that initially digitized, prepared, and shared their data, not knowing that it is a follow-on process, not only a one-time upload.

For full implementation, countries need to have a basic NSDI strategy defined and institutions responsible for their geospatial data need to have mini-strategies that cover all business processes, from data collection to data sharing. According to the authors, countries that already have prepared metadata can fully establish the necessary services for accessing geospatial data within a short period of time. The roadmap to achieve it

depends on new policies and new technological developments. However, since the legal obligation is present, in order to improve the current situation, INSPIRE NCPs can reinforce their role. Better coordination and collaborations are needed between all NSDI subjects. INSPIRE and related NSDIs should evolve from complex and highly specialized geospatial data frameworks, where legal obligations are enforced by strict technical specifications, to flexible, agile, sustainable, and data-driven ecosystems [9].

## 4. Discussion and Conclusions

The INSPIRE directive reacted to the growing need for sharing and exchanging interoperable spatial data and services across Europe for policy-making and to the fact that the spatial data situation in Europe was considered "one of fragmentation, gaps in availability, duplication of information collection and problems of identifying, accessing or using data that is available" [25]. One of the main benefits of the INSPIRE directive is an improvement in the functioning of the public administration at all levels by facilitating administrative access to geospatial information. It includes availability through the metadata and accessibility of spatial data and spatial data services. The overarching vision for a European SDI has not changed since the inception of the INSPIRE directive, which is to promote cross-border data sharing and to put in place easy-to-use, transparent, interoperable spatial data services, which are used in the daily work of environmental and other policy makers and policy implementers across the EU at all levels of governance as well as businesses, science, and citizens. With the deadlines of the INSPIRE implementation in 2021, the implementation of the directive has entered a final phase. However, looking back on the past implementation cycle, we have to state that implementation is delayed, and gaps are evident.

The analysis made within this paper had the aim to investigate and evaluate at what level the obligations stated in the INSPIRE roadmap were fulfilled. Regarding the subject of the special issue of this scientific journal, we focused on two water-related spatial data themes: hydrography and hydrogeology (sub-theme of spatial data theme geology). To be coherent with formal INSPIRE monitoring and reporting, we used several indicators to assess availability and accessibility of hydrography and hydrogeology data (see Section 2.4). The analysis included metadata records available through the INSPIRE Geoportal Thematic Viewer and the resource browser to assess the availability and accessibility of hydrography and hydrogeology datasets. In addition, we investigated view and download services more deeply to compare numbers of declared services on the INSPIRE Geoportal with the ones we could really access through the provided links (URLs).

Based on the calculated INSPIRE monitoring indicators for hydrography and hydrogeology, our analysis showed the discrepancies between countries as well as inside of each country. Both data themes should be available and accessible in all countries, but it is not the case. There are countries with zero metadata records. This should not be the case since every country should have hydrography and hydrogeology data. We also noticed that some countries make declarations of large numbers of available services and datasets, but in reality, provide small amounts of them. In contrast, the countries that declare a very small number of available datasets and services generally provide them all. Furthermore, we analyzed the access to each of the declared spatial datasets and found a significant gap regarding the declared data: only 52% of the countries had really accessible and downloadable datasets or services in the hydrography theme, while 48% equaled zero. The status of INSPIRE implementation is very heterogeneous across the EU, with some countries performing well and others still lagging. This is not only with hydrography and hydrogeology data, but also in other thematic areas. It is fully in line with the results of the INSPIRE evaluation done in 2021 [26].

Looking back at the past implementation cycle of INSPIRE, we should state that implementation is still delayed and gaps are evident. INSPIRE is more convincing in theory than in practice. Obviously it must be changed. We should not forget that INSPIRE is a legal instrument and not only a voluntary initiative.

In this article, we did not aim to discuss or answer how to close the INSPIRE implementation gaps but rather to reflect current facts and shortcomings. It could be a possible topic for future research. INSPIRE requires a delicate balance among public, private, and personal interests while taking into account the complex interplay among technological, legal, economic, and institutional issues in achieving such a balance [27]. Collaborative partnerships and joint efforts of many organizations are needed to establish and maintain modern NSDIs in MS and EFTA countries as dynamically evolving infrastructures.

Emerging technologies (i.e., AI, APIs, big data, and others) together with current and envisaged policy initiatives (e.g., open data directive, European data strategy, Green-Data4All) should be the drivers for the future INSPIRE evolution.

The reasons for the lack of hydrological geospatial data vary, from financial problems, lack of decisions at the management level, lack of institutional guidelines and adequately trained staff, capacity, etc. Lack of geospatial data, their possible poor quality, incompleteness, and unavailability can cause serious problems, primarily in business processes and, subsequently, in the field of economic indicators in almost all parts of society.

All institutions that take care of hydrological geospatial data should have a clear hierarchy of responsibilities and implement technical guidelines in the area of the IN-SPIRE directive. Good and quality hydrological data, their availability, integration, and interoperability with other datasets, clear implementation rules, and ultimately a reliable authoritative source, are the bases for the main pillars of each country, regarding the economy, environment, health, saving lives, agriculture and food production, tourism, employment, and other branches, and each state should take care of them and strengthen capacities at all levels, as well as provide methods of implementation for their daily use.

**Author Contributions:** Conceptualization, V.C. and D.M.; methodology, V.C., S.Š., O.B.O. and D.M.; validation, V.C. and D.M.; formal analysis, V.C., S.Š., O.B.O. and D.M.; investigation, S.Š. and O.B.O.; data curation, S.Š. and O.B.O.; writing—original draft preparation, V.C. and O.B.O.; writing—review and editing, V.C., S.Š., O.B.O. and D.M.; visualization, V.C., S.Š. and O.B.O.; supervision, V.C. and D.M. All authors have read and agreed to the published version of the manuscript.

**Funding:** The APC was funded by the scientific project *3D geoinformation for the purpose of irrigation canal modeling* from the University North, Croatia, with PhD Sanja Šamanović as a project lead.

**Data Availability Statement:** All the data used in the analysis of this research are publicly available through INSPIRE Geoportal.

**Conflicts of Interest:** The authors declare no conflict of interest.

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
