# Peer review of "Availability and Accessibility of Hydrography and Hydrogeology Spatial Data in Europe through INSPIRE"

_water, doi:10.3390/w14091499_

Round 1

Reviewer 1 Report

I want to congratulate the authors. I think your paper is important to the science community. The paper describes the EU initiative "INSPIRE" (Infrastructure for Spatial Information in Europe), the database platform to establish a European Union Spatial Data Infrastructure (SDI) for the purposes of the EU’s environmental policies and policies or activities which have an impact on the environment. A directive came into force on 15 May 2007, with full implementation in every Member State required by 2021.

The paper describes the current state of implementation and usability according to the FAIR data principles and draws conclusions for the future use and further development of the INSPIRE infrastructure and the INSPIRE geoportal.

In my view, this paper is important to draw more attention to this initiative and that this tool can be better used in the future.

Data analysis (Part 3) is for as a practicing user not very clear formulated since I am unfortunately rather overwhelmed by the indicators described scattered in the text. Therefore, the tables are not very informative for me.

Page 10/11: The mentioned DSi1.1-DSi1.5 indicators are not clearly explained and therefore confusingly. It would be nice to add a clarification for example, as a suffix in Table 1 or in an own table.

Author Response

Thank you for your revision. Please find enclosed our feedback on your comments.

Kind regards

Authors

Reviewer 2 Report

The current manuscript discusses the progress of INSPIRE – a platform to share EU spatial data. The writing is OK but the reviewer had the following comments:

1) The introduction should include not only information about INSPIRE. As stated in lines 18-19, availability and accessibility of public spatial data in the EU have been always an issue due to policies, formats, licenses, prices, etc. INSPIRE should not be the first trial to create a pan-EU platform of spatial data. The introduction should include past efforts in this regard. The second part (“INSPIRE”) of the manuscript mentioned a little bit of this but was not comprehensive at all.

2) Figure 2: the blank blocks can be eliminated or reduced in size. It looks as if some texts are missing.

3) Figure 3: texts in the figure are too small to read

4) Figure 4 is included in the manuscript without being mentioned in the texts.

5) Line 142: Please make sure the entry point is depicted in Figure 3.

6) Lines 182-183: please avoid a paragraph containing only one sentence.

7) Figure 9: a legend should be provided for the abbreviations of nations in the figure. Similar measures should be done to other tables/figures using abbreviated country names.

8) Figures 9-13: titles are not required in the figures since captions are available.

9) The reviewer had the impression that this manuscript is a “midterm report” on the progress of INSPIRE. The goal of INSPIRE has not been achieved. There are gaps in data availability. The reviewer does not believe that reporting issues of an incomplete data platform has any contribution to the scientific community. The reviewer urges the authors to wait until the completion of INSPIRE to resubmit the manuscript.

Author Response

Thank you for your revision. Please find enclosed our feedback on your valuable comments.

Kind regards

Authors

Round 2

Reviewer 2 Report

Thanks for providing revisions to issues found in the previous round of review. The issues (wrong figure numbers, for example) found were only a glimpse of all possible issues. The authors should be responsible for removing all similar issues from the manuscript.

On the very last issue, the reviewer is still not totally convinced. The authors argued that pointing out errors is important in encouraging institutions to solve problems. The reviewer agrees with that. However, a more constructive way of criticism is to provide ways for improvements. The reviewer expects to see a thorough description of how the missing parts can be filled, and also the timeline/roadmap to do it.

Author Response

We would like to thank you for your review and comments given regarding our paper. We have tried to follow all your suggestions and regarding your comments we have improved our paper.

Reviewer#2 comment#1: Thanks for providing revisions to issues found in the previous round of review. The issues (wrong figure numbers, for example) found were only a glimpse of all possible issues. The authors should be responsible for removing all similar issues from the manuscript.

Authors: Thank you for your comment. We checked the whole article again to avoid these types of errors.

Reviewer#2 comment#2: On the very last issue, the reviewer is still not totally convinced. The authors argued that pointing out errors is important in encouraging institutions to solve problems. The reviewer agrees with that. However, a more constructive way of criticism is to provide ways for improvements. The reviewer expects to see a thorough description of how the missing parts can be filled, and also the timeline/roadmap to do it.

Authors: Thank you for your comment. We have added our proposals for improvement in the Chapter 3. Line 399 – Line 450.

The way how to improve the availability and accessibility of data is not easy. In this article we touched only two INSPIRE data themes. Based on it, it is clear that the overall implementation is delayed. Nevertheless, there are new movements: current evaluation of INSPIRE Directive and way forward, Implementation of the Open Data Directive together with implementing act on High Value Data Sets, etc.. Technology is changing on the daily basis which also influence the way how the infrastructure is/will be implemented in the future.
